# Emotion-body connection dispositions modify the insulae-midcingulate effective connectivity during anger processing

Viridiana Mazzola[1,2]*, Giampiero Arciero[3,4], Leonardo Fazio[5], Tiziana Lanciano[6], Barbara Gelao[5], Alessandro Bertolino[5], Guido Bondolfi[1,2]

**1** Department of Psychiatry, Liaison Psychiatry, University Hospitals of Geneva, Geneva, Switzerland, **2** Department of Psychiatry, Faculty of Medicine, University of Geneva, Geneva, Switzerland, **3** Department of Psychiatry, University Hospitals of Geneva, Geneva, Switzerland, **4** Institute of Post-Rationalist Psychology IPRA, Rome, Italy, **5** Department of Basic Medical Sciences, Neuroscience and Sense Organs, University of Bari, Bari, Italy, **6** Department of Education, Psychology, Communication, University of Bari, Bari, Italy

* Viridiana.Mazzola@hcuge.ch

**Data Availability Statement:** The University of Bari Aldo Moro is not allowed to share data beyond the terms accepted by participants in the informed consent form at the time of recruitment. The

## Abstract

The link between anger and bodily states is readily apparent based on the autonomic and behavioral responses elicited. In everyday life angry people react in different ways, from being agitated with an increased heart rate to remaining silent or detached. Neuroimaging evidence supports the role of mid-posterior insula and midcingulate cortex/MCC as key nodes of a sensorimotor network that predominantly responds to salient stimuli, integration of interoceptive and autonomic information, as well as to awareness of bodily movements for coordinated motion. However, there is still a lack of clarity concerning how interindividual variability in bodily states reactions drives the connectivity within these key nodes in the sensorimotor network during anger processing. Therefore, we investigated whether individual differences in body-centered emotional experience, that is an active (inward prone) or inactive (outward prone) emotion-body connection disposition, would differently affect the information flow within these brain regions. Two groups of participants underwent fMRI scanning session watching video clips of actors performing simple actions with angry and joyful facial expressions. The whole-brain group-by-session interaction analysis showed that the bilateral insula and the right MCC were selectively activated by inward group during the angry session, whereas the outward group activated more the precuneus during the joyful session. Accordingly, dynamic causal modeling analyses (DCM) revealed an excitatory modulatory effect exerted by anger all over the insulae-MCC connectivity in the inward group, whereas in the outward group the modulatory effect exerted was inhibitory. Modeling the variability related to individual differences in body-centered emotional experience allowed to better explain to what extent subjective dispositions contributed to the insular activity and its connectivity. In addition, from the perspective of a hierarchical model of neurovisceral integration, these findings add knowledge to the multiple ways which the insula and MCC dynamically integrate affective and bodily aspects of the human experience.

responsible ethics committee at the University of Bari Aldo Moro is: Comitato etico indipendente | Azienda Ospedaliero-Universitaria Consorziale Policlinico | Piazza Giulio Cesare 11, 70124 Bari | e-mail: comitatoetico@policlinico.ba.it | phone: +39 080 559 3399. Data access request can be directed to Prof. Alessandro Bertolino (alessandro. bertolino@uniba.it) and Dr. Giulio Pergola (giulio. pergola@uniba.it).

**Funding:** The authors received no specific funding for this work.

**Competing interests:** The authors have declared that no competing interests exist.

## Introduction

Our bodily states dynamically change along with our emotional experiences. According to the kind of the emotions in question, our bodies are differentially affected in terms of autonomic reactivity and readiness to sets of possible actions [1,2]. This becomes evident also when we consider anger, where the link between emotions and bodily states is readily apparent based on the autonomic and behavioral responses elicited [3,4]. Indeed, in everyday life, angry people may begin to yell and experience the urge to throw or break things, they may feel hot and have an increased and rapid heart rate. Others react in a different way. When angry they become silent, even sarcastic, they withhold without perceiving neither heat nor an increase in heart rate, as if this anger-bodily states connection were almost not active. Accordingly, it is not surprising that anger plays a role as a risk factor and trigger for acute cardiovascular events in some but not all persons [5–14]. The psychology of emotions describes anger experience variability in terms of type and relevance of triggers (who is involved and the situation in question), autonomic involvement, duration and regulation, as well as behavioral outcome [15]. However, such an approach does not lead to a fully understanding of individual differences in bodily states reactions in response to anger experience. At the brain level, the insula cortex may constitute a link between cognitive, sensorimotor, and social-emotional systems in human behavior [16]. Neuroimaging research has provided robust evidence supporting the hub function of insula and its connections in dynamically integrating affective and bodily aspects of the inner and outer, as well as our sense of limb ownership and limb movement awareness [17–19]. Several results about insula connectivity showed how the mid-posterior insula in particular is linked to motor and somatosensory cortices [20,21]. Precisely, the mid-posterior insula together with the midcingulate cortex/MCC monitor the ongoing sensory context in order to select the behavioral responses via skeletomotor control and orient our body according to the salient stimuli [22–25]. Further, our previous fMRI study showed that the right mid-posterior insula and its connectivity play a key role while facing others' anger [26]. Thus, the mid-posterior insulae and MCC are together key nodes of a sensorimotor network that predominantly respond to salient stimuli, integrate interoceptive and autonomic information, and increase our awareness of bodily movements for coordinated motion [17,20,21,27–29]. Nevertheless, there is still a lack of clarity about how individual differences in emotion-body connection would differently drive the insula-MCC causal relation during emotional processing.

Given the main role played by the bodily states in response to anger, individual differences in emotion-body connection during emotional experiences might be considered as an important source of heterogeneity that challenges our understanding of the anger-bodily states connection and its neural signatures. Therefore, the question arises as to whether an active or inactive attitude in emotion-body connection would differently engage the insulae-MCC connectivity when facing anger situations. To address this question, we selected two groups of participants according to their attitude to focus primary on their bodily activations (inward prone) or on contexts or people (outward prone) to make sense of the ongoing situation when emotionally involved [30–33]. Our concept of dispositional affective styles emphasizes the need to account for the way in which each person, in dealing with others and the different circumstances of everyday life, feels situated in the environment. Within this perspective, two general dispositional affective styles can be defined: primarily based on basic emotions (inward disposition) or primarily based on emotions which are co-perceived through others (non-basic emotions) (outward disposition) [2,34,35]. On one hand, inward subjects tend to be more body-centered, i.e. viscerally aware, more sensitive in the detection of changes in bodily states occurring during emotional experience [32,33]. On the other hand, during emotional

experience outward subjects tend to be more context-centered, i.e. externally aware, focusing primarily towards contexts, people or rules and norms [32,33].

In the present study, all the participants faced angry and joyful situations during the fMRI scanning session. They watched two runs of video clips of an actor and an actress performing a daily act like grasping an object with different emotional expressions (neutral, joyful, or angry) [26,36]. One run included only neutral and joyful expressions, while the other featured only neutral and angry expressions. We compared the contextual modulation of connection strengths within the insulae-MCC connectivity elicited by the emotional situations in the two groups using dynamic causal modeling (DCM) analyses. For this purpose, we created an architecture that served as our base model to test how the two emotional acts were more likely to exert an excitatory or an inhibitory effect within our target connection. Then, we obtained statistical estimates of which model offered the optimum balance between simplicity and fit to the data using Fixed Effects Bayesian Model Selection family inference analysis (FFX BMS) [37], followed by the model parameter estimates analysis Bayesian Parameters Averaging (BPA) (see Methods). The resting state data were acquired before the task in order to check for any non task-related difference between groups in the mid-posterior insulae-MCC connectivity. Spectral DCM analyses of resting state data were performed according to the winning model structure [38].

Based on the evidence mentioned, we hypothesized that the mid-posterior insulae-MCC connectivity would be differently engaged by the priors for one's emotion-body connection disposition during emotional processing. In particular, we expected that anger vs joy would have a stronger excitatory modulatory effect on connectivity within the mid-posterior insulae-MCC in the inward disposition than in the outward one.

## Methods

### Participants

The sample was drawn from two larger cohorts of people who participated in a previous psychometric study according to the In-Out dispositional affective style questionnaire/IN-OUT DASQ and the MRI scanning criteria [33]. Indeed, the current participants were classified as higher inward or higher outward prone depending on their highest scores given to the IN-OUT DASQ subscales, the Self-centric engagement for the inwardness and Other-centric engagement for the outwardness [33]. In addition, a semi-structured interview was administered independently by two trained investigators (GA, TL) who were blind to each other's results. Two groups of fifteen participants each were enrolled. Due to technical problems, a participant of the outward group was excluded. In order to better balance the two groups we excluded one participant of the inward group matched for gender and age. The final two groups were composed by fourteen participants each (Inward group: 7 females; mean age 27.36; standard deviation [SD] 2.92, range = 22–33; Outward group: 8 females; mean age 26.50; standard deviation [SD] 3.5, range = 22–34). Exclusion criteria included a history of drug or alcohol abuse, previous head trauma with loss of consciousness, pregnancy, and any significant medical or psychiatric conditions as evaluated with the Structured Clinical Interview (SCID). All the participants had more than 16 years of schooling. After the semi-structured interview, participants were asked to describe the way they usually get angry, namely, what is the trigger (situations), the level of involvement of bodily states in terms of higher-medium-low intensity, how long their anger lasts and how it ends (duration and regulation). All the participants completed the State Trait Anger Expression Inventory (STAXI) [39] in order to control for their general tendency to experience anger. Before the scanning session, each participant also completed the State-Trait Anxiety Inventory (STAI) [40] to evaluate their

current state of anxiety in addition to the other questionnaires. The present study was approved by the Comitato Etico Indipendente Locale of the Azienda Ospedaliera "Ospedale Policlinico Consorziale" in Bari, Italy. Informed written consent was obtained from all participants before participation. All methods were performed in accordance with the relevant guidelines and regulations.

## Self-report questionnaires analyses

Questionnaire data were analyzed using SPSS 23 (Inc., 2009, Chicago, USA). Mann-Whitney U tests between groups were performed to test for the group differences effect on DASQ, STAXI, and between males and females on the STAXI measures to test the gender effect. We reported the exact significance level (two-tailed). In order to test for any significant correlation between DASQ, STAXI, we performed Spearman's rho correlation analyses. We reported the exact significance level (two-tailed).

## fMRI task

The fMRI task employed was described in detail in other previous studies [26,36]. Briefly, the functional MRI session consisted of two successive scanning runs with an event-related design. Each run included one emotion (joy or anger), plus the neutral facial expression, and consisted of four experimental view conditions. All visual stimuli consisted of video clips of 1.3 second. There were 4 different video conditions in each run, showing the following actions: the trunk with an arm grasping an object on a table (acting alone), a person with a neutral facial expression grasping an object on a table (neutral acting), a person with a joyful or angry facial expression grasping an object on a table (joyful acting in the joyful run or angry acting in the angry run), and a joyful or angry dynamic facial expression without any acting action (joyful face in the joyful run or angry face or in the angry run). The recording and editing of videos were made using the Blue Screen technique in order to superimpose on the same trunk different emotional facial expressions. Thereby, we investigated the brain activity elicited by the observation of someone acting in angry and joyful situations, while keeping action kinematics constant. Two professional actors, a female and a male, were enrolled as models for the videos. The everyday objects to be grasped were put on a table, e.g. phone, pen, keys, bottle, cup, and glass. To circumvent any motor interference, we used a passive viewing task and participants were instructed to remain still without performing any movement, and to avoid any imitation or mental imagery of the actions shown, and to carefully look at the video clips in order to get involved. The presentation order of the two runs was counterbalanced across subjects. In each scanning run, there were 160 visual stimuli presented in random order, with an average 1810 ms interstimulus interval (ISI). 48 additional null events, each lasting 2700 ms, contributed to randomly jitter the stimulus onsets. Total scanning run time was about 19 minutes.

## fMRI data acquisition and analyses

Three-dimensional images were acquired using a T1-weighted SPGR sequence (TR/TE/NEX = 25 s/3 ms/1; flip angle 6˚; matrix size 256×256; FOV 25×25 cm) with 124 sagittal slices (1.3 mm thick, in-plane resolution of 0.94×0.94). Resting state and task-related fMRI data were acquired on a 3T GE (General Electric, Milwaukee, WI) MRI scanner with a gradient-echo echo planar imaging (EPI) sequence and covered 26 interleaved axial slices (5 mm thick, 1mm gap), encompassing the entire cerebrum and the cerebellum (TR 2 s; FOV 24 cm; matrix, 64 x 64, a voxel size of 3.75x3.75x5 mm). A total of 150 EPI volume images were acquired for the resting state, whereas for each scan of the task, a total of 285 EPI volume images.

## Preprocessing

Data were preprocessed and analyzed using statistical parametric mapping SPM12 (Wellcome Department of Cognitive Neurology, London, UK), implemented in MatLab R2014b (MathWorks[TM]). A fixed-effect model at a single-subject level was performed to create images of parameter estimates, which were then entered into a second-level random-effects analysis. For each subject, functional images were first slice-timing corrected, using the middle slice acquired in time as a reference, and then spatially corrected for head movement, using a least-squares approach and six-parameter rigid body spatial transformations. High-resolution anatomical T1 images were coregistered with the realigned functional images to enable anatomical localization of the activations. The two runs were then entered as multiple sessions as implemented in SPM12. Structural and functional images were spatially normalized into a standardized anatomical framework using the default EPI template provided in SPM12, based on the averaged-brain of the Montreal Neurological Institute and approximating the normalized probabilistic spatial reference frame of Talairach and Tournoux [http://brainmap.org/icbm2tal/]. Functional images were spatially smoothed with a three-dimensional Gaussian filter (10mm full-width at half-maximum). The time series was temporally filtered to eliminate contamination from slow drift of signals (high-pass filter, 128 s) and corrected for autocorrelations using the AR(1) model in SPM12.

## General linear model

We performed two parallel but identical statistical analyses on the functional data for the whole-brain and cerebellar normalized images. Four event-types were defined per subject per scanning run, corresponding to each condition of interest. In the joyful run, the conditions of interest were: acting alone, neutral acting, joyful acting, joyful face. In the angry run, the conditions were: acting alone, neutral acting, angry acting, angry face. Eight contrast images corresponding to these conditions from individual participants were entered at the second level into a repeated-measures 2 (groups) x 2 (sessions) x 4 (conditions) ANOVA (flexible factorial design implemented in SPM12). We reported regions that survived a threshold of $P < 0.05$ cluster-level FWE corrected, cluster size Ke = 8, as implemented in SPM12. Cerebral MNI coordinates were converted to the Talairach coordinate system by icbm2tal [http://brainmap.org/icbm2tal/]. Anatomic and Brodmann areas labeling of cerebral activated clusters was performed with the Talairach Daemon database [http://www.talairach.org/] and SPM Anatomy Toolbox [41].

## Cerebellar normalization

We used a separate normalization process for data from the cerebellum. The registration between individuals and MNI space is suboptimal in the cerebellum when using a standard whole-brain normalization process [42]. Because cerebellum vary relatively little between individuals compared with the cortical landmarks used for whole-brain normalization, it is possible to achieve a much better registration by normalizing the cerebella separately. Moreover, precise spatial registration is important because cerebellar structures are small compared to cortical structures. To this aim, we used the SUIT toolbox [42] for SPM12 allowing us to normalize each individual's structural scan to an infratentorial template, and then used the resulting deformation maps to normalize the cerebellar sections of each person's functional images. The SUIT toolbox has the additional advantage that coordinates can be adjusted from MNI space to the corresponding coordinates on the unnormalized Colin-27 brain, which is described anatomically in a cerebellar atlas. We used this feature to identify anatomical regions within the cerebellum.

## Dynamic causal modeling

**VOI extraction.** Volumes of interest (VOIs) were defined bilaterally in the insula and in the MCC based on the sensorimotor network [22,25]. The centers of the VOIs were identified as the peaks of sensorimotor independent component map. To identify these coordinates, a spatial independent component analysis (ICA) was performed in the resting state on a joint group with all the participants using the Group ICA of fMRI Toolbox (GIFT) (http://icatb. sourceforge.net/). GICA3 was used for back-reconstruction type and 20 components were extracted. The resulting independent component maps were verified by visual inspection to identify the sensorimotor network. The regions included the right insula (centered at MNI space 42, -18, 4), the left insula (centered at MNI space -42, -18, 2), the right MCC (centered at MNI space 4, 4, 28), and the left MCC (centered at 0, −2, 34). For each subject, the VOIs were defined as spheres centered at those coordinates mentioned above with a 6 mm radius. This procedure ensures that cross-spectral density and of task-related DCM analyses were performed on the same VOIs coordinates identified as a temporally coherent network.

**Specification of model architecture.** All the DCM analyses were performed with the DCM12 routine implemented in SPM12. We used dynamic causal modeling (DCM) to examine between-groups differences in insula-MCC effective connectivity, i.e. the impact that activity in one region exerts over another [43]. More specifically, the DCM explains changes in neuronal population dynamics as a function of the network's connectivity (endogenous connectivity) and regional effects in terms of the changing patterns of connectivity among regions according to the experimentally induced contextual modulation of connection strengths. Accordingly, we examined the mutual influences within this brain regions involved both in the angry vs joyful session using DCM. Our DCM design matrices comprised a regressor modeling of the sensory input (all conditions in each session), and a regressor modeling of the effect of the emotional acting condition. We grouped into three families all the models according the following connections: RINS↔MCC↔LINS, RINS↔LINS↔MCC as a right top-down family, LINS↔MCC↔RINS, LINS↔RINS↔MCC as a left top-down family, and MCC↔RINS↔LINS, MCC↔LINS↔RINS as a bottom-up family. For each architecture we constructed 17 different models with bidirectional connections allowing experimental modulation on all paths and regions, resulting 102 possible DCMs per emotional session per side of MCC (Fig 1). In all models, the first region received all the conditions of the session as a driving input. In order to answer our research questions, we varied the regions and connections that were affected by the emotional (angry and joyful) acting across the models (Fig 1).

**Model comparison.** Inference on family structure was performed using Fixed Effects family inference analysis (FFX BMS) [37]. The model's Free Energy, F, a lower bound of the model's log-evidence, accounting for model complexity as well as data fit, was used to compare the likelihood of the different models to explain the data. Relative log-evidences, or differences in F, were converted into model posterior probabilities, p, indicating that the respective family has a probability p of being the best family/model explaining the data amongst all considered. Evidence was "strong" if $p > 0.95$, which stands for a difference in F greater than 3, and "positive" if $0.75 < p < 0.95$, which stands for a difference of F between 1 and 3 [37,44]. Secondly, inference on the optimal model parameters was performed. The structure of the connectivity model was assumed to be the same for both sequences and a FFX analysis of the model parameter estimates was performed using Bayesian Parameters Averaging (BPA) [45–47].

Since spectral DCM has been found to be more accurate and more sensitive to group differences compared to stochastic DMC [38], DCM of cross spectral density analyses of the winning model were performed to control for any significant differences in connectivity strength between groups at resting state [38]. A spectral DCM with all endogenous connectivity

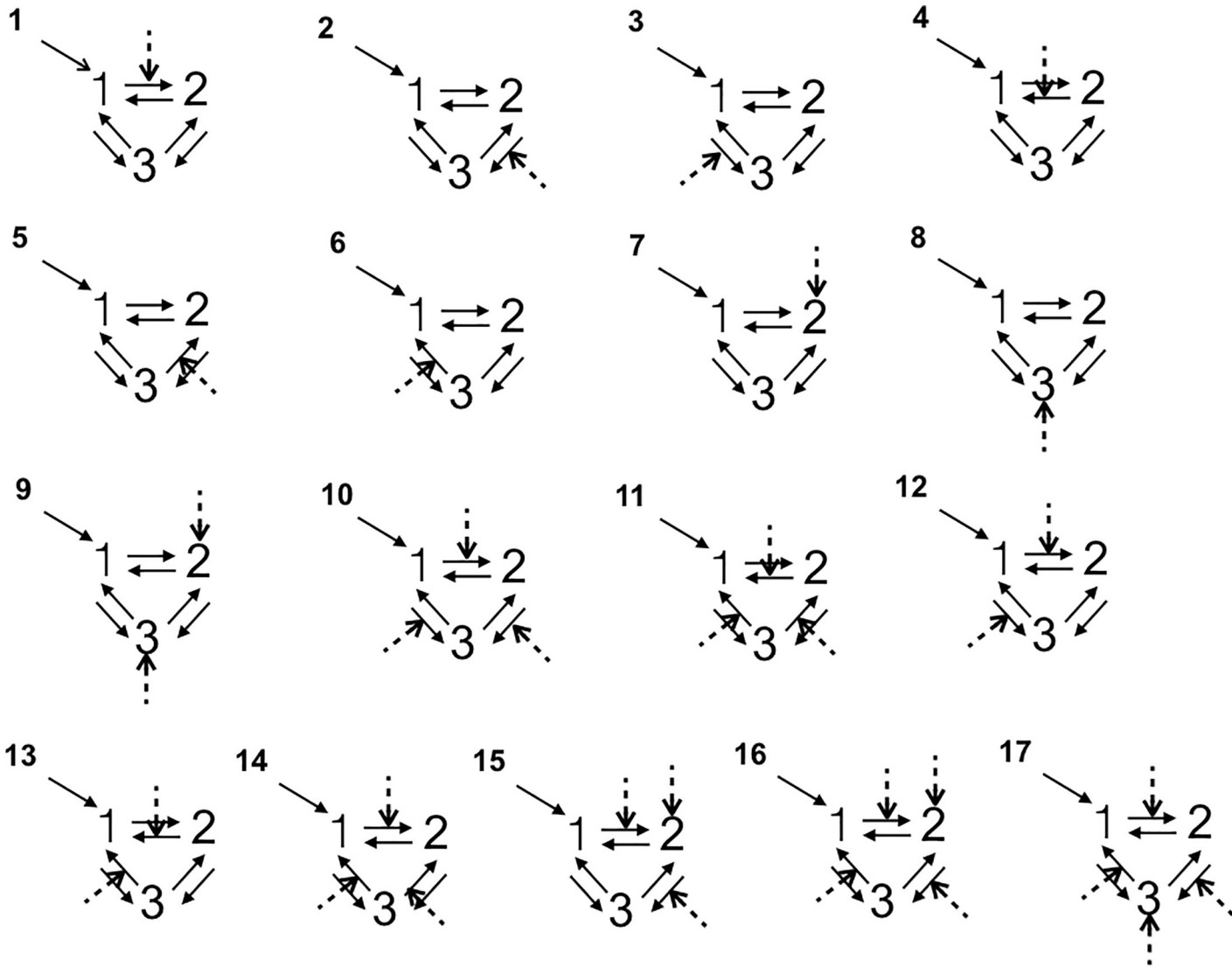

**Fig 1. The models tested with Bayesian model selection.** Numbers 1, 2, 3 refer to the regions of interest. The dashed lines indicate the modulatory influences of emotional acting on connectivity among right insula (rINS), left insula (lINS), and right/left MCC (MCC).

specified (full model) was built for each participant. The BPA approach was adopted to obtain model parameters for each group, separately. Then, a two sample T-test was performed on group parameters to check for any significant difference.

## Results

### Self-report questionnaires

The two groups significantly differed in the Self-centric engagement/SCE scale scores. Indeed, the inward group scored significantly higher (W 141, p<0.05, mean difference 6, SE 2.76, d' 0.646) (Table 1). This result was consistent with the semi-structured interview classification. According to the STAXI assessment of the experience, expression and control of the anger, the two groups showed only a different tendency to control the anger expression. Indeed, the inward group scored higher on the Anger control scale (W 139.5, p<0.05, mean difference 3,

**Table 1. Self-report questionnaires.**

| | INWARD | | OUTWARD | |
|---|---|---|---|---|
| Scale | Mean | SD | Mean | SD |
| Self-centric engagement | 27.21 | 5.01 | 22.50 | 9.019 |
| Other-centric engagement | 29.64 | 6.08 | 27.50 | 7.623 |
| State-anger | 15.43 | 6.11 | 12.57 | 3.5 |
| Trait-anger | 20.50 | 4.64 | 22.28 | 5.45 |
| Trait-anger-temperament | 6.71 | 1.86 | 6.86 | 2.85 |
| Trait-anger-response | 9.85 | 2.62 | 11.36 | 2.37 |
| Anger-in | 19.36 | 2.73 | 18.50 | 5.27 |
| Anger-out | 14.57 | 2.85 | 14.92 | 4.99 |
| Anger control | 26.64 | 3.71 | 22.43 | 6.32 |
| Anger expression | 23.28 | 6.14 | 27.00 | 9.56 |

SE 1.95, d' 0.813). Consistently, the inwards' SCE scores correlated negatively with the inward Anger control (r = -0.68 p<0.001), and positively with the Anger expression (r = 0.71 p<0.001), and the Trait-anger-temperament (r = 0.76 p<0.001). Taken together, these results suggest that the inward dispositional affective style revealed a different profile in anger expression. Indeed, the inward participants showed a tendency to be more angered and, accordingly, to control their anger expression. These results were consistent with the usual way of getting angry described by all the inward participants. Actually, at the apex of their own anger, they reported to perceive a general involvement of their body (in terms of the urge to act also against the person/situation, increasing transpiration and heart beating, feeling hot) that lasted for a time interval of between 30 minutes and 2 hours according to the relevance of the episode. On the other hand, the outward participants referred that they usually did not get angry but rather irritated, at the peak they did not report to perceive any significant change in bodily states, and the peak lasted from 2 to 15 minutes in the most relevant episodes.

## Neuroimaging Results

**Whole-brain fMRI results.** Across sessions, the inward group activated the right mPFC, the bilateral ACC, the left MCC, and left insula more, as well as the right VII lobule crus I and II (P<0.05 FWE corrected) (Table 2, Fig 2A). On the other hand, the outward participants showed a greater involvement of visual areas, right precuneus, right precentral, right parahippocampus and the right anterior cerebellum (P<0.05 FWE corrected) (Table 2, Fig 2B). Across groups, the angry session elicited higher activity in bilateral superior frontal gyrus and the right posterior insula (P<0.05 FWE corrected) (Table 2). In contrast, no activation reached the statistical significance in the reverse contrasts.

To address whether the processing of anger impacts on the activity of the network of salience and action link system, we carried out a group x session interaction. According to our predictions, we found greater activation in the salience-action link system during the angry session than the joyful one. Indeed, the bilateral mid-posterior insula and the right MCC were engaged by the angry session along with the left caudate, right hippocampus, left parahippocampus, and left precuneus (P<0.05 FWE cluster-level corrected) (Table 3). Namely, the activity in the bilateral mid-posterior insula and the right MCC was mainly driven by the inward group. On the other hand, the right mid-posterior insula and the left precuneus activations were enhanced by the joyful session in the outward group.

Group x emotional acting interaction also revealed a differential activity. Indeed, the angry session elicited activity in the left claustrum, the right putamen and caudate, right STG, right

**Table 2. Whole-brain general linear model main effects analyses, P<0.05 FEW. Inward group > Outward group across sessions.**

**MNI coordinates**

| Region | x | y | z | Ke | Z Scores |
|---|---|---|---|---|---|
| Right Medial Prefrontal Cortex BA10 | 24 | 44 | 10 | 276 | 5.9 |
| Right ACC BA32 | 14 | 34 | 8 | | 5.89 |
| Right IPL BA40 | 32 | -42 | 46 | 61 | 5.56 |
| Left ACC BA32 | -20 | 40 | 10 | 201 | 5.43 |
| Left MCC BA24 | -10 | -16 | 36 | | |
| Left Insula BA13 | -34 | 16 | 18 | 20 | 5.38 |
| Left Claustrum | -24 | 22 | 0 | 20 | 5.23 |
| Right Superior Temporal Gyrus BA22 | 54 | -32 | 2 | 26 | 5.22 |
| L Pallidum | -16 | 0 | -2 | 17 | 5.1 |
| Left Cingulate Cortex BA32 | -14 | 16 | 36 | 61 | 5.04 |
| Right VIIa crusII | 42 | -54 | -51 | 699 | 5.36 |
| Right VIIa crusI | 48 | -48 | -35 | 363 | 5.81 |
| Left VIIIa lobule | -40 | -48 | -61 | 352 | 4.86 |
| **Outward group > Inward group across sessions** | | | | | |
| Right Lingual Gyrus BA18 | 16 | -72 | -2 | Inf | 332 |
| Left Lingual Gyrus BA18 | -10 | -80 | 2 | 222 | 6.24 |
| Right Middle Occipital gyrus BA37 | 38 | -64 | 6 | 128 | 6.09 |
| Right Fusiform FG2 | 40 | -70 | -20 | 66 | 5.89 |
| Right Amygdala | 18 | 2 | -16 | 48 | 5.7 |
| Right Calcarine Cortex BA18 | 4 | -76 | 16 | 49 | 5.48 |
| Right Precentral Gyrus BA4 | 24 | -22 | 66 | 9 | 4.89 |
| Right Middle Frontal Gyrus BA46 | 34 | -2 | 58 | 11 | 4.82 |
| Right Parahippocampal Gyrus BA28 | 16 | -12 | -14 | 8 | 4.78 |
| Right IV lobule | 30 | -38 | -23 | 236 | 4.33 |
| **Angry > Joyful sessions across groups** | | | | | |
| Left Superior Frontal Gyrus BA10 | -20 | 50 | -2 | 178 | 4.82 |
| Right Superior Frontal Gyrus BA10 | 18 | 50 | -6 | 133 | 4.68 |
| Right Insula Ig2 | 30 | -24 | 12 | 86 | 4.56 |

paracentral lobule, and right thalamus (P<0.05 FWE cluster-level corrected) (Table 3). Specifically, the activity in the right mid-insula was mainly driven by the inward group as well as the right MCC (Fig 2C and 2D).

Consistent with our a priori hypothesis, these results support the view that facing others' anger enhances activity within the mid-posterior insulae-MCC network in the inward participants.

## Effective connectivity

Next, we applied DCM analyses to investigate the causal architecture of insulae-MCC effective connectivity that may account for group differences in processing of others' emotions. Among the models constructed with the right MCC, the right top-down family was the winning in the inward group during the angry session with a posterior probability (Pp) of .99. The optimal model showed a positive modulation by angry acting on the forward connection from the right mid-posterior insula both in the left mid-posterior insula and the right MCC, and on the backward connection from the right MCC to the left mid-posterior insula, as well as a direct slight negative effect on the right MCC (Fig 3A) (Pp = .99). This winning model reveals noteworthy findings on the information flow when inward participants processed angry people.

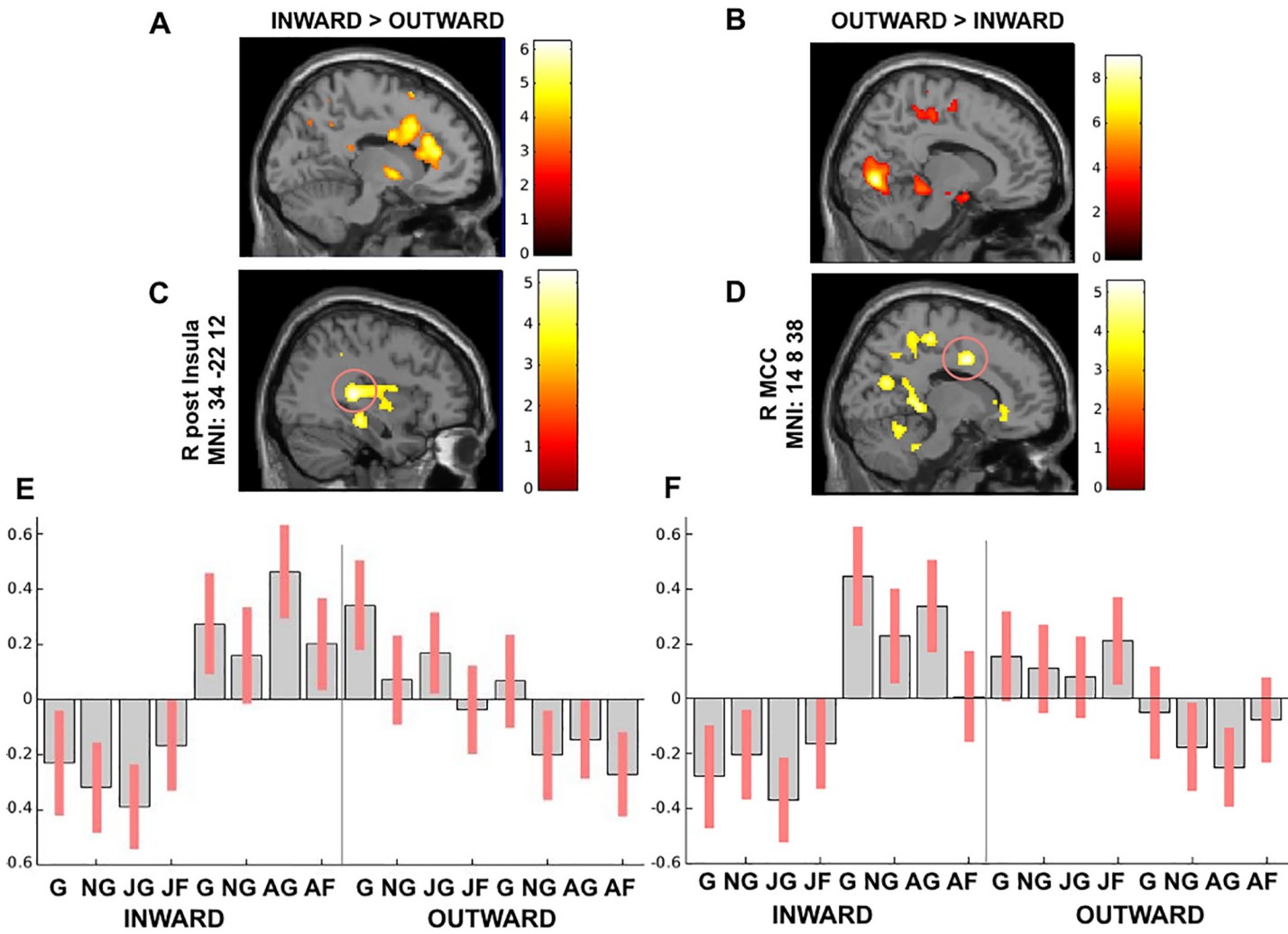

**Fig 2.** **(A)** Inward group > Outward group across sessions and **(B)** Outward group > Inward group across sessions, whole-brain analysis between groups P<0.05 FWE; **(C-D)** The peak signal changes that occurred in the right posterior insula and right MCC in groups x emotional acting interaction results (P<0.05 FWE cluster-level corrected). **(E-F)** Contrast estimates and 90% confidence intervals in the right posterior insula and in the right MCC presented for visualization purpose.

First, it revealed the central role played by the right insular as input when facing an anger situation. Indeed, angry acting had an excitatory modulatory effect from the right mid-posterior insula both to the left mid-posterior insula and the right MCC. Secondly, the right MCC played the role of the output node in the winning model. Moreover, the right MCC backward connection to the left insula was positively modulated by angry acting.

On the other hand, the winning family in the outward group was the left top-down one (Pp = .99) in the angry session, showing that the input role was played by the left insula (Fig 3D). The optimal model showed a negative modulation by angry acting on the backward connection from the right MCC to the left insula and from the right insula to the left insula, as well as a direct negative effect on the right MCC (Pp = .99). This winning model revealed a different information flow in the outward group. Indeed, the main modulatory effect of the angry acting was to inhibit the input region, i.e. the left insula, and the right MCC.

Among the models constructed with the left MCC, the left top-down family won in the inward group when facing anger (Pp = 1) (Fig 3B). The optimal model revealed that the angry

**Table 3. Whole-brain general linear model interaction analyses, P<0.05 FWE. Groups x sessions.**

| | MNI coordinates | | | | |
| --- | --- | --- | --- | --- | --- |
| **Region** | **x** | **y** | **z** | **Ke** | **Z Scores** |
| Right Hippocampus CA1 | 36 | -28 | -16 | 4491 | 5.76 |
| Right ACC BA32 | 12 | 30 | -10 | | 5.21 |
| Right Insula BA13 | 34 | -22 | 12 | 436 | 4.69 |
| Left Insula BA13 | -32 | -24 | 8 | 114 | 4.34 |
| Left Caudate | -20 | 26 | 0 | 899 | 5.48 |
| Left Parahippocampal Gyrus BA37 | -36 | -36 | -6 | 1816 | 5.45 |
| Right MCC BA24 | 14 | 2 | 38 | 947 | 462 |
| Right Calcarine Cortex BA31 | 16 | -64 | 16 | 166 | 4.5 |
| Right Precuneus BA7 | 14 | -62 | 42 | 323 | 4.12 |
| **Groups x emotional acting interaction** | | | | | |
| Left Claustrum | -22 | 26 | 0 | 448 | 5.07 |
| Right Superior Temporal Gyrus BA41 | 36 | -32 | 6 | 865 | 5.01 |
| Right Insula Ig1 | 34 | -24 | 10 | | |
| Right Parahippocampal Gyrus CA1 | 34 | -26 | -16 | | |
| Left Parahippocampal Gyrus BA37 | -36 | -36 | -4 | 686 | 4.31 |
| Right Putamen | 28 | -18 | 2 | | |
| Right MCC BA24 | 14 | 0 | 38 | 333 | 5 |
| Right Paracentral lobule BA6 | 12 | -30 | 54 | 460 | 4.61 |
| Right Precuneus BA5 | 10 | -44 | 58 | | |
| Right Lingual Gyrus BA30 | 16 | -40 | -2 | 278 | 4.57 |
| Right Thalamus | 10 | -28 | -2 | | 3.57 |
| Right Caudate | 2 | 10 | 0 | 271 | 4.47 |

acting positively modulated the backward connection from the right insula to the left one, and negatively from the left MCC to the left insula, with a direct slight negative effect on the right insula (Pp = .99). This model showed that there was not any reciprocal influence between the right insula and left MCC in the inward group. In the outward group, the winning family was the right top-down, but no one model reached the probability threshold above chance.

The joyful session elicited an opposite information flow compared to the angry one in the inward group. Indeed, among the models with the right MCC, the winning family was the bottom-up (Pp = 1). The optimal model showed an inhibitory modulatory effect exerted by joyful acting from the RMCC to the right insula and a positive one to the left insula, as well from the left insula to the right one, and a direct inhibitory effect on the left insula (Pp = .81) (Fig 3C). In the outward group the winning family was the right top-down (Pp = 1), whereas no one model related to the joyful session reached the probability threshold above chance in the outward group. Among the left MCC models, in both the groups the winning family was the left top-down (Pp = 1) and no one model reached the probability threshold above chance.

DCM cross-spectral analyses revealed that the two groups did not differ significantly concerning the strength of connections in the winning models during the resting-state.

## Discussion

Given the interindividual variability in bodily states reactions when facing anger, our research question concerned whether the angry vs joyful context differently affected the mid-posterior insulae-MCC connectivity according to individual differences in emotion-body connection. To this aim, we enrolled participants who had two distinct priors for one's emotion-body

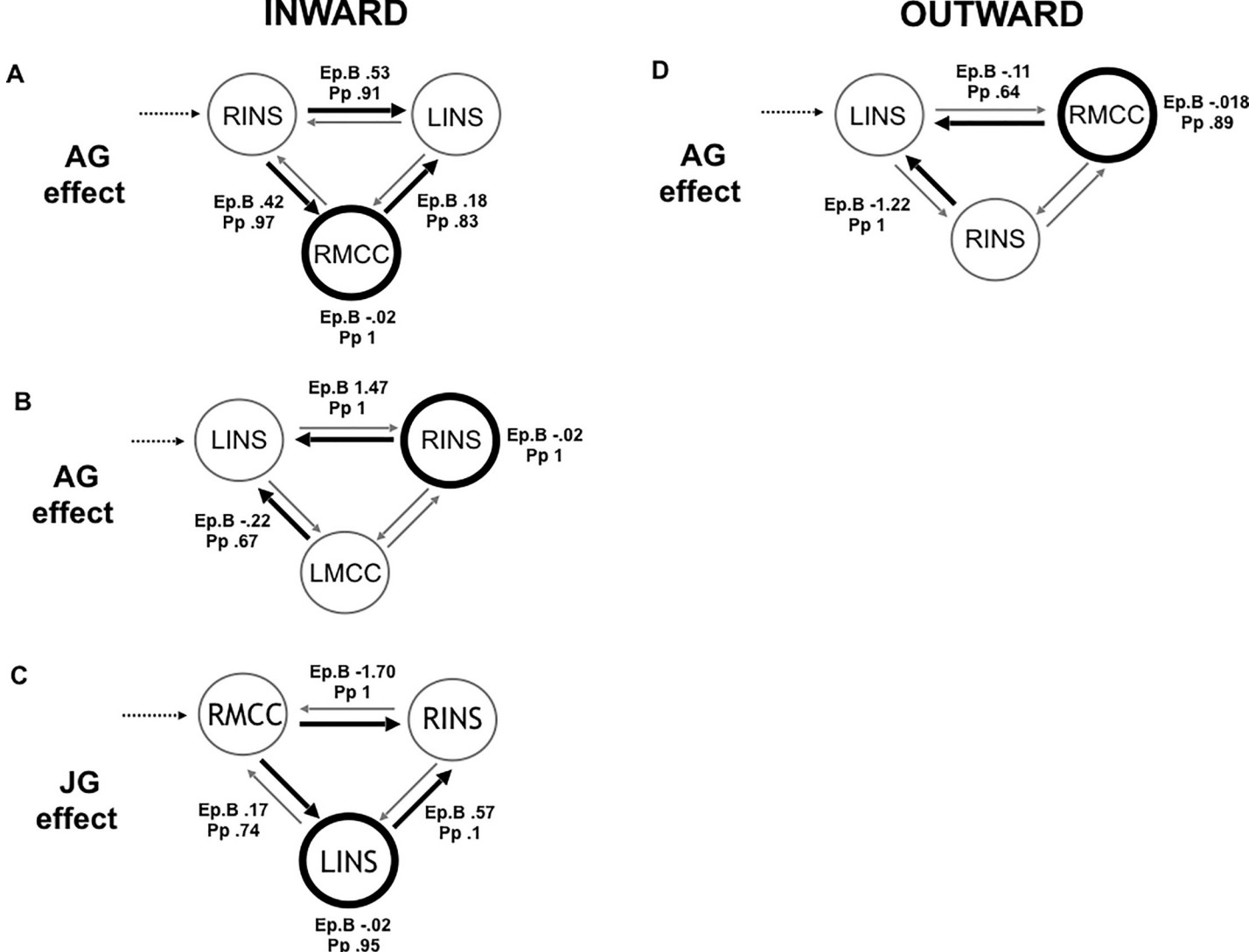

**Fig 3. The winning models.** Black dotted lines indicate input into the system by all conditions; grey lines indicate fixed connectivity; black bold lines and circles indicate the modulatory effect of emotional acting. Ep.B: connection strength index; Pp: posterior probability.

connection disposition. Indeed, the inward disposition focuses more on bodily state changes when they are affectively engaged, while the outward disposition focuses on the external world and people. During the fMRI scanning session, the participants watched video clips of actors doing an ordinary action like grasping objects from a table in joyful and angry contexts.

The DCM analyses here employed allowed us to provide a more systematic and nuanced directional depiction of the connectivity between the mid-posterior insulae and the MCC in our two groups. The present findings let us point out two major aspects. First, the winning model in the inward group revealed that angry acting spread an excitatory modulatory effect all over the forward connections from the right mid-posterior insula and on the right MCC. On the other hand, the DCM model that better explained the modulatory effect of the angry acting in the outward group depicted an opposite information flow. It spread an inhibitory modulatory effect all over the backward connections to the input, i.e. the left mid-posterior insula, and on the right MCC. These findings confirmed our expectation about the different

involvement of these key nodes within the sensorimotor network in our groups when processing anger situation.

Second, in the inward group the angry acting effect modulated more selectively the right-sided ipsilateral MCC-insula connectivity, whereas in the outward group the contralateral one, that is right MCC-left insula. This reveals that the right-sided ipsilateral insula-MCC causal relation had been prioritized by the inward disposition when processing anger-related situation, unraveling the neural mechanism underlying the inward emotion-body connection. Consistently, under joyful acting the modulatory effect was reversed in the inward group. Indeed, the joyful acting spread from the right MCC an inhibitory effect on the right insula. Indeed, this depiction of connectivity showed a shift in the emotion-body connection activation in the inward group.

The whole-brain analyses also supported our hypothesis about a different affective engagement between the two groups. Indeed, the inward group showed across sessions a general involvement of more limbic areas, whereas the outward group more visual areas. Accordingly, the group-by-session interaction analysis showed that the bilateral insula and the right MCC were selectively activated by the inward group during the angry session. On the other hand, the activity in the precuneus was mainly driven by the outward group. This result is consistent with the inward-outward affective engagement as revealed by our previous study on pain empathy [32].

Taken together, these findings reveal that an active or inactive emotion-body connection disposition modulates differently the causal relation within the insulae-MCC in response to angry and joyful situations, highlighting a different neural implementation of emotional processing. Clearly, in people prone to be more body-centered, when exposed to others' anger, the information flow started from a hierarchically higher cortical level, the right mid-posterior insula, and ended in a lower subordinate cortical level, the right MCC. This right-sided insula-MCC downward connectivity may be referred to the anger-brain-body connection described by these participants at a peak of their angry behavioral level as an 'urge to act'. Indeed, both these regions are directly involved in skeletomotor body orientation and response selection [17,21,27]. In addition, as a visceromotor limbic cortex, the right MCC is directly implied as the output in the 'urge to act' feeling elicited by anger, after having received excitatory input from the right mid-posterior insula [48–50]. Furthermore, both the mid-posterior insula and the MCC are key regions within the central autonomic network (CAN) [51], playing a direct role in sympathetic regulation [52]. Assuming the lateralized hypothesis of insular autonomic control, according to which the right insula elicits more sympathetic cardiovascular reactions, e.g. tachycardia and hypertension, whereas the left side elicits more the parasympathetic reactions, e.g. bradycardia and hypotension [53–57], these results could be consistent with a general increased sympathetic response elicited by anger in inward participants. Unfortunately, the lack of physiological measures concerning autonomic activity acquired during the scanning session is a limitation of the present study that cannot let us further address this interpretation.

Consistent with our previous fMRI study, the present study confirmed a primary role of the right insula when processing angry situation [26]. In addition, modeling the variability related to individual differences in body-centered emotional experience allowed us to better explain the subjective affective dispositions contribution to the insular activity and connectivity. This is an important contribution if we consider to what extent psycholgical factors may adversely affect other medical conditions by, for instance, influencing the underlying pathophysiology to precipitate or exacerbate symptoms, as in the case of chronic pain. As a matter of fact, the insula-MCC connectivity plays the primary role as a gate for nociceptive hypersensitivity, as well as the intrinsic connectivity between insula-MCC at the resting state is also altered in

patients affected by chronic pain [58,59]. Moreover, in line with the concept of precision medicine, according to which the prevention and treatment strategies has to take individual variability into account, the present findings may be useful in personalizing anger management in several clinical domains where anger effects are known to be detrimental [9,12,13,60].

To sum up, the depiction of the causal relation between the insulae-MCC drawn out by angry vs joyful situations in the inward group was consistent with their behavioral attitude as revealed by both the STAXI scores and the description of their own anger experience. Indeed, what emerged as a group effect was their tendency to get angry and consequently to control their anger expression more, intertwined with a lasting bodily involvement that clearly resembles a sympathetic-like response. It is worth noting that there was no significant difference between the two groups in high-low trait anger scores. Thus, these differences in attitude toward anger between groups, outlined by the semi-structured interview and supported by the questionnaires, were observed also at a neural level within the sensorimotor network connectivity. It implies that the sensorimotor network, or at least part of it, is dynamically involved according to the ongoing emotional situation and the kind of affective disposition. At the same time, we acknowledge that due the sample size further investigations are required.

In conclusion, the results show a different neural implementation of anger vs joyful processing within the insulae-MCC effective connectivity in accordance with individual differences in emotion-body connection dispositions. Indeed, a prior disposition to be more external context- or body-centered during emotional experiences differently affects the information flow within these key nodes of the sensorimotor network. This leads us to remark that the dispositional affective styles here considered open to a much more fine-grained understanding of the variability of the anger experience, unveiling other facets of the anger-brain-body connection phenomenon. Finally, in the perspective of a hierarchical model of neurovisceral integration [51,61,62], the dimensions of inward and outward emotion-body connection dispositions as a priors add knowledge to help understand the multiple ways of the insula and its connections to dynamically integrate affective and bodily states of the human experience.

## Author Contributions

**Conceptualization:** Viridiana Mazzola, Giampiero Arciero.

**Data curation:** Viridiana Mazzola, Giampiero Arciero, Leonardo Fazio, Tiziana Lanciano, Barbara Gelao.

**Formal analysis:** Viridiana Mazzola.

**Investigation:** Leonardo Fazio, Tiziana Lanciano, Barbara Gelao.

**Methodology:** Viridiana Mazzola.

**Resources:** Alessandro Bertolino.

**Supervision:** Alessandro Bertolino, Guido Bondolfi.

**Writing – original draft:** Viridiana Mazzola, Giampiero Arciero.

**Writing – review & editing:** Viridiana Mazzola, Giampiero Arciero.

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
