## [Decision Letter · Decision Letter 0]

19 Aug 2019

PONE-D-19-19373

How the insulae-midcingulate connectivity changes as a function of individual differences in emotion-body connection during anger processing.

PLOS ONE

Dear Dr. Mazzola,

Thank you for submitting your manuscript to PLOS ONE. After careful consideration, we feel that it has merit but does not fully meet PLOS ONE’s publication criteria as it currently stands. Therefore, we invite you to submit a revised version of the manuscript that addresses the points raised during the review process.

We would appreciate receiving your revised manuscript by September 20, 2019. To enhance the reproducibility of your results, we recommend that if applicable you deposit your laboratory protocols in protocols.io, where a protocol can be assigned its own identifier (DOI) such that it can be cited independently in the future. For instructions see: http://journals.plos.org/plosone/s/submission-guidelines#loc-laboratory-protocols

We look forward to receiving your revised manuscript.

Kind regards,

Mukesh Dhamala, Ph. D.

Academic Editor

PLOS ONE

**Journal Requirements**

**Comments to the Author**

1. Is the manuscript technically sound, and do the data support the conclusions?

Reviewer #1: Partly

2. Has the statistical analysis been performed appropriately and rigorously? 

Reviewer #1: No

3. Have the authors made all data underlying the findings in their manuscript fully available?

Reviewer #1: Yes

4. Is the manuscript presented in an intelligible fashion and written in standard English?

Reviewer #1: Yes

5. Review Comments to the Author

Reviewer #1: Summary: In this manuscript, authors identified the role of insula and midcingulate cortex (MCC) to explain the individuals differences in body-centered emotional experience. Authors also used effective connectivity approach to establish the link between anger and effective connectivity between insula and MCC. Overall, manuscript provides a strong ‘Introduction’ section, but weaker ‘Methods’ section, especially in explaining and using DCM approach - which may further have an impact on ‘Results’ and ‘Discussion’ sections.

I have following suggestions for the authors:

Title: I would suggest the authors to modify the title and replicate the title with the main finding from the manuscript.

Introduction: Overall, the ‘Introduction’ section is very well written and provides the relevant background in detail. I have following suggestions:

- Line 76: Please define the abbreviation MCC (midcingulate cortex).

- Line 111: Please clarify which approach was used for DCM analysis – whether it was Bayesian Model Selection (BMS) or Bayesian Parameter Average (BPA) or BMS followed by BPA. I think it’s only BMS which provides optimal model as mentioned in the following sentence. Also, please define the abbreviation BPA here.

- Line 112: Typo: ‘resting state’ should be replaced by resting state data’.

- Lines 113-114: Please define the term ‘effective connectivity’ before using this term in the Introduction.

- Line 114: Its confusing whether authors used traditional DCM technique or spectral DCM approach, or both - one of task-based condition and the other for resting-state. If authors used both, then please clarify the rationale behind using two approaches rather than using traditional DCM for both.

- Line 115: It’s not clear how “according to model” phrase is used by the authors to define expected engagement of specific connectivity because the whole idea behind defining a model or model space is to test all the hypotheses defined by each model. Authors provided useful literature in earlier parts of the ‘Introduction’, so I would suggest to set up the hypothesis and expected findings based on literature and then define the model space to test the hypothesis. I would recommend to rearrange and modify the last paragraph of ‘Introduction’ section.

Methods:

- Lines 129-130: Rationale behind excluding data of one subject from second group (due to technical difficulties in first group) is not clear. I believe that you do not need to always have equal number of participants to perform DCM.

- I was wondering if the parameters such as ‘years of education’ and ‘ethnicity/race’ were accounted/recorded while comparing two groups.

- Line 175: Please define the units of scanning parameters (TR/TE/NEX).

- Line 180: Slice thickness of 5 mm for fMRI data is very large in general as well as compared to T1w data of 1.3 mm thickness. How did the authors make sure that the thicker slices did not have negative impact on activation and effective connectivity results?

- Lines 232-234: Please clarify if the ROIs co-ordinates reported here are in MNI space or TAL space.

- Lines 238-239: Please revise these sentences.

- Lines 240-241: Please revise the rationale here, because DCM not only explains regional effects in terms of modulation of connection strengths, but also modulation of regions themselves as well as in absence of modulation of connections and regions.

- Line 250: Idea behind using either left MCC or right MCC is not strong. Did the authors perform any tests to figure out why one side was preferred over the other? Second, if authors already defined ROIs with 6 mm radius with peak coordinates as the center, then I am not sure why authors didn’t use the full area for DCM? Third, in earlier section, authors mentioned that spherical ROIs were used so that identical ROIs can be used for DCM analysis, so I am not sure how that idea is supported by the using MCC – either for the left hemisphere or the right.

- Overall, the model space for DCM analysis is clear. However, the ‘Method’ section is missing the important details about spectral DCM (as well as traditional DCM). Because spectral DCM is a one of the latest techniques implemented by Razi et. al., therefore I would suggest the authors to include all the necessary details so that the readers can benefit. Appropriate citations of previous spectral DCM papers would be beneficial.

Results:

- It’s not clear whether the results accounted for covariates such ‘age’ and ‘gender’. One of the recently published studies showed sex differences (https://onlinelibrary.wiley.com/doi/full/10.1002/jnr.24504) in the limbic network and impact of age on different cortical networks (https://www.frontiersin.org/articles/10.3389/fnagi.2017.00412/full). Moreover accounting for the effects of ‘age’ and ‘gender’ are also recommended for spectral DCM analysis.

- Line 317: It’s not clear how did the authors calculate and compare posterior probability (Pp) of family of models? That could be the reason authors are getting perfect values of Pp (= 1) for some of the families. Please clarify the idea behind this approach. Did the authors compare Pp of 103 models altogether and tried to figure out the best model among 103?

- Resolution for Figure 2 (bar plots) is very low. Purpose of this figure is not clear. Also, please define the abbreviations used in Figure 3 (e.g. Ep.B).

6. PLOS authors have the option to publish the peer review history of their article (what does this mean?). If published, this will include your full peer review and any attached files.

Reviewer #1: No

---

## [Author Response · Author response to Decision Letter 0]

18 Sep 2019

PONE-D-19-19373

Emotion-body connection dispositions modify the insulae-midcingulate effective connectivity during anger processing.

PLOS ONE 

Dear Dr Dhamala,

first of all, we thank the reviewer for his/her useful comments.

We have answered all the questions in reviewing the manuscript. 

Respectfully yours,

Viridiana Mazzola, PhD, PsyD

Reviewer #1:

We changed the title.

Line 76: We defined the MCC abbreviation.

Line 111: We clarified the approach we used.

Line 112: We did it.

Lines 113-114: We did it.

Line 114: We clarified this point.

Line 115: We rearranged the last paragraph of the Introduction according to the reviewer’s suggestion.

Lines-129-130: We Due to the small size of the groups, we preferred to better balance the number of the participants within each group in order to avoid any potential confounding.

We added the years of schooling we recorded. Since in the present study we did not collect genetic data, we did not consider the variable “ethnicity”. On the other hand, the culture (for example western vs eastern) we agree that it can have an impact on the emotional processing. All the participants have the same cultural background, i.e. Italians.

Line 175: We defined the units.

Line 180: We used both these standard fMRI sequences as we did in our previous studies employing the same fMRI task in order to keep a continuity (Mazzola et al. 2013, 2016). We thought that both the two sequences were sufficiently optimal to balance the sensitivity of fMRI to stimulus-correlated motion and spatial resolution for cortical regions (not subcortical structures) according to our regions of interest. Nevertheless, we agree to reconsider this issue for the next fMRI studies. We thank the reviewer for this point. 

Lines 232-234: We did it.

Lines 240-241: We corrected this point.

Line 250: We decided to use either left or right MCC according to a degree of lateralization observed in insula-MCC connectivity and salience network (Cauda et al, 2011, 2012; Taylor et al. 2009). 

We added all the necessary details about the spectral DCM.

Cauda, F., D'Agata, F., Sacco, K., Duca, S., Geminiani, G., Vercelli, A., (2011). Functional connectivity of the insula in the resting brain. Neuroimage 55, 8–23.

Cauda, F., Costa, T., Torta, D. M. E., Sacco, K., D’Agata, F., Duca, S., … Vercelli, A. (2012). Meta-analytic clustering of the insular cortex. Characterizing the meta-analytic connectivity of the insula when involved in active tasks. NeuroImage, 62(1), 343–355. http://doi.org/10.1016/j.neuroimage.2012.04.012

Taylor, K. S., Seminowicz, D. A., & Davis, K. D. (2009). Two systems of resting state connectivity between the insula and cingulate cortex. Human Brain Mapping, 30(9), 2731–2745. http://doi.org/10.1002/hbm.20705

Line 317: We clarified it. 

We improved the resolution of figure 2. 

We defined the abbreviations in Fig. 3.

---

## [Decision Letter · Decision Letter 1]

24 Oct 2019

PONE-D-19-19373R1

Emotion-body connection dispositions modify the insulae-midcingulate effective connectivity during anger processing.

PLOS ONE

Dear Dr Mazzola,

Thank you for submitting your manuscript to PLOS ONE. After careful consideration, we feel that it has merit but does not fully meet PLOS ONE’s publication criteria as it currently stands. Therefore, we invite you to submit a revised version of the manuscript that addresses the points raised during the review process.

We would appreciate receiving your revised manuscript by Dec 08 2019 11:59PM. To enhance the reproducibility of your results, we recommend that if applicable you deposit your laboratory protocols in protocols.io, where a protocol can be assigned its own identifier (DOI) such that it can be cited independently in the future. For instructions see: http://journals.plos.org/plosone/s/submission-guidelines#loc-laboratory-protocols

We look forward to receiving your revised manuscript.

Kind regards,

Mukesh Dhamala, Ph. D.

Academic Editor

PLOS ONE

Additional Editor Comments (if provided):

I would like to ask the authors to respond to this round of reviews.

Reviewers' comments:

Reviewer's Responses to Questions

**Comments to the Author**

1. If the authors have adequately addressed your comments raised in a previous round of review and you feel that this manuscript is now acceptable for publication, you may indicate that here to bypass the “Comments to the Author” section, enter your conflict of interest statement in the “Confidential to Editor” section, and submit your "Accept" recommendation.

Reviewer #1: (No Response)

2. Is the manuscript technically sound, and do the data support the conclusions?

Reviewer #1: Partly

3. Has the statistical analysis been performed appropriately and rigorously? 

Reviewer #1: No

4. Have the authors made all data underlying the findings in their manuscript fully available?

Reviewer #1: Yes

5. Is the manuscript presented in an intelligible fashion and written in standard English?

Reviewer #1: Yes

6. Review Comments to the Author

Reviewer #1: I feel that authors have not fully addressed/implemented most of the major comments/concerns. In addition, it has not been described whether their revised approach based on my previous comments will have any impact of the results/discussion. Below I am outlining my previous comments once again regarding this manuscript:

- Line 111: Please clarify which approach was used for DCM analysis – whether it was Bayesian Model Selection (BMS) or Bayesian Parameter Average (BPA) or BMS followed by BPA. I think it’s only BMS which provides optimal model as mentioned in the following sentence. Also, please define the abbreviation BPA here.

- Lines 113-114: Please define the term ‘effective connectivity’ before using this term in the Introduction.

- Line 114: Its confusing whether authors used traditional DCM technique or spectral DCM approach, or both - one of task-based condition and the other for resting-state. If authors used both, then please clarify the rationale behind using two approaches rather than using traditional DCM for both.

- Line 115: It’s not clear how “according to model” phrase is used by the authors to define expected engagement of specific connectivity because the whole idea behind defining a model or model space is to test all the hypotheses defined by each model. Authors provided useful literature in earlier parts of the ‘Introduction’, so I would suggest to set up the hypothesis and expected findings based on literature and then define the model space to test the hypothesis. I would recommend to rearrange and modify the last paragraph of ‘Introduction’ section.

- For : “Lines 129-130: Rationale behind excluding data of one subject from second group (due to technical difficulties in first group) is not clear. I believe that you do not need to always have equal number of participants to perform DCM”, again I am not sure about the potential confounds and selection of excluded subject from DCM analysis. For DCM analysis, there is no need to make the sample size equal.

- Lines 240-241: I am still not sure about the rationale of this study based on which spectral DCM analysis was used.

- Line 250: Idea behind using either left MCC or right MCC is not completely addressed. Did the authors perform any tests to figure out why one side was preferred over the other? Second, if authors already defined ROIs with 6 mm radius with peak coordinates as the center, then I am not sure why authors didn’t use the full area for DCM? Third, in earlier section, authors mentioned that spherical ROIs were used so that identical ROIs can be used for DCM analysis, so I am not sure how that idea is supported by the using MCC – either for the left hemisphere or the right.

- I am not sure where did the authors include details/ideas about spectral DCM?

- It’s not clear whether the results accounted for covariates such ‘age’ and ‘gender’. One of the recently published studies showed sex differences (https://onlinelibrary.wiley.com/doi/full/10.1002/jnr.24504) in the limbic network and impact of age on different cortical networks (https://www.frontiersin.org/articles/10.3389/fnagi.2017.00412/full). Moreover accounting for the effects of ‘age’ and ‘gender’ are also recommended for spectral DCM analysis.

- Line 317: It’s not clear how did the authors calculate and compare posterior probability (Pp) of family of models? That could be the reason authors are getting perfect values of Pp (= 1) for some of the families. Please clarify the idea behind this approach. Did the authors compare Pp of 103 models altogether and tried to figure out the best model among 103?

7. PLOS authors have the option to publish the peer review history of their article (what does this mean?). If published, this will include your full peer review and any attached files.

Reviewer #1: No

---

## [Author Response · Author response to Decision Letter 1]

10 Dec 2019

PONE-D-19-19373R1

Emotion-body connection dispositions modify the insulae-midcingulate effective connectivity during anger processing.

PLOS ONE 

Dear Dr Dhamala,

we thank again the reviewer for his/her useful comments.

We have answered all the questions in reviewing the manuscript. 

Respectfully yours,

Viridiana Mazzola, PhD, PsyD

We are sorry we did not fully addressed the reviewer’s comments. We believe that thanks to the reviewer’s comments our manuscript has been improved.

Reviewer #1: - Line 111: Please clarify which approach was used for DCM analysis – whether it was Bayesian Model Selection (BMS) or Bayesian Parameter Average (BPA) or BMS followed by BPA. I think it’s only BMS which provides optimal model as mentioned in the following sentence. Also, please define the abbreviation BPA here.

As written in Model comparison section, inference on family model structure was performed using Fixed Effects Bayesian Model Selection family inference analysis (FFX BMS) (Penny et al., 2010), then a FFX analysis of the model parameter estimates was performed using Bayesian Parameters Averaging (BPA)(Acs and Greenlee, 2008, Garrido et al., 2007; Neumann and Lohmann, 2003).

We briefly added in introduction section.

Reviewer #1:- Lines 113-114: Please define the term ‘effective connectivity’ before using this term in the Introduction.

We did write only ‘mid-posterior insulae-MCC connectivity’.

Reviewer #1:- Line 114: Its confusing whether authors used traditional DCM technique or spectral DCM approach, or both - one of task-based condition and the other for resting-state. If authors used both, then please clarify the rationale behind using two approaches rather than using traditional DCM for both.

With respect to resting state fMRI, spectral DCM has been found to be more accurate and more sensitive to group differences compared to stochastic DMC (Razi et al 2015). On the other hand, for task-related fMRI the most commonly and validated methods to investigate context-related perturbations of effective connections between brain regions is a basic deterministic DCM approach irrespective of group differences (Friston et al., 2003). Accordingly, we used both the approaches.

We added this rational in the methods section (see line 271.)

Reviewer #1:- Line 115: It’s not clear how “according to model” phrase is used by the authors to define expected engagement of specific connectivity because the whole idea behind defining a model or model space is to test all the hypotheses defined by each model. Authors provided useful literature in earlier parts of the ‘Introduction’, so I would suggest to set up the hypothesis and expected findings based on literature and then define the model space to test the hypothesis. I would recommend to rearrange and modify the last paragraph of ‘Introduction’ section.

At line 115 of the current manuscript we cannot find this sentence. We did already changed according to the reviewer’s comment.

Reviewer #1:- For : “Lines 129-130: Rationale behind excluding data of one subject from second group (due to technical difficulties in first group) is not clear. I believe that you do not need to always have equal number of participants to perform DCM”, again I am not sure about the potential confounds and selection of excluded subject from DCM analysis. For DCM analysis, there is no need to make the sample size equal.

The rationale to equal the two samples size was to balance the design and avoid any potential criticism and difficulties of interpretation of results about unbalanced design. We did perform also other analyses besides the DCM analysis, in which case an unbalanced design would have been a potential problem and a choice hard to justify in our opinion.

The participant not included in the second group was the one selected according to the inclusion criteria to match participants, that is, same age and gender of the excluded one due to technical problem. 

Reviewer #1:- Lines 240-241: I am still not sure about the rationale of this study based on which spectral DCM analysis was used.

See the answer to the previous question about Line 114.

Reviewer #1:- Line 250: Idea behind using either left MCC or right MCC is not completely addressed. Did the authors perform any tests to figure out why one side was preferred over the other? Second, if authors already defined ROIs with 6 mm radius with peak coordinates as the center, then I am not sure why authors didn’t use the full area for DCM? Third, in earlier section, authors mentioned that spherical ROIs were used so that identical ROIs can be used for DCM analysis, so I am not sure how that idea is supported by the using MCC – either for the left hemisphere or the right.

Line 255: we cancelled the sentence that was ambiguous. We got the MNI coordinates of the MCC, as well as of the mid-posterior insula, from the ICA analysis. The resulting independent component maps had the both sides. Then, we included both side of the MCC separately in each model in order to respect the criteria of the parsimony in constructing DCM models and to be more precise in testing the close integration between MCC and mid-posterior insula. Indeed previous results showed a degree of lateralization observed in insula-MCC connectivity and salience network (Cauda et al, 2011, 2012; Taylor et al. 2009). The DCM analysis figured out which was the preferred one.

We utilized the same VOIs coordinates obtained from ICA analysis both for the resting-sate DCM analysis and the task-related DCM analysis.

Reviewer #1:- I am not sure where did the authors include details/ideas about spectral DCM?

Please, see lines 271-75

Reviewer #1:- It’s not clear whether the results accounted for covariates such ‘age’ and ‘gender’. One of the recently published studies showed sex differences (https://onlinelibrary.wiley.com/doi/full/10.1002/jnr.24504) in the limbic network and impact of age on different cortical networks (https://www.frontiersin.org/articles/10.3389/fnagi.2017.00412/full). Moreover accounting for the effects of ‘age’ and ‘gender’ are also recommended for spectral DCM analysis.

We did not covariate for age and gender in order to account for group differences. Modeling for within and between-group differences in terms of age and gender effect is of interest (thanks for the references!), but beyond the hypothesis of the present study. Accordingly, the rs-fMRI analysis was conceived only to check for any significant differences between groups in the regions of interest not due to task-related activity.

Reviewer #1:- Line 317: It’s not clear how did the authors calculate and compare posterior probability (Pp) of family of models? That could be the reason authors are getting perfect values of Pp (= 1) for some of the families. Please clarify the idea behind this approach. Did the authors compare Pp of 103 models altogether and tried to figure out the best model among 103?

The Pp reported in the manuscript of the winning family and model was calculated by the DCM12 routine implemented in SPM12 and saved in the BMS.mat file. We grouped the 102 models in three families as described in the paragraph “Specification of model architecture”, line 250. 

The Pp reported was an approximation of .9999. We changed “Pp 1” with “Pp .99”.

---

## [Decision Letter · Decision Letter 2]

15 Jan 2020

Emotion-body connection dispositions modify the insulae-midcingulate effective connectivity during anger processing.

PONE-D-19-19373R2

Dear Dr. Mazzola,

We are pleased to inform you that your manuscript has been judged scientifically suitable for publication and will be formally accepted for publication once it complies with all outstanding technical requirements.

With kind regards,

Mukesh Dhamala, Ph. D.

Academic Editor

PLOS ONE

Additional Editor Comments (optional):

The authors have appropriately addressed all the comments raised by the reviewer. I can now recommend it for publication.

Reviewers' comments:

Reviewer's Responses to Questions

**Comments to the Author**

1. If the authors have adequately addressed your comments raised in a previous round of review and you feel that this manuscript is now acceptable for publication, you may indicate that here to bypass the “Comments to the Author” section, enter your conflict of interest statement in the “Confidential to Editor” section, and submit your "Accept" recommendation.

Reviewer #1: All comments have been addressed

2. Is the manuscript technically sound, and do the data support the conclusions?

Reviewer #1: Yes

3. Has the statistical analysis been performed appropriately and rigorously? 

Reviewer #1: Yes

4. Have the authors made all data underlying the findings in their manuscript fully available?

Reviewer #1: Yes

5. Is the manuscript presented in an intelligible fashion and written in standard English?

Reviewer #1: Yes

6. Review Comments to the Author

Reviewer #1: Authors have fully addressed all of my concerns and questions, and have appropriately revised the manuscript.

7. PLOS authors have the option to publish the peer review history of their article (what does this mean?). If published, this will include your full peer review and any attached files.

Reviewer #1: No

---

## [Editor Report · Acceptance letter]

3 Feb 2020

PONE-D-19-19373R2 

Emotion-body connection dispositions modify the insulae-midcingulate effective connectivity during anger processing. 

Dear Dr. Mazzola:

I am pleased to inform you that your manuscript has been deemed suitable for publication in PLOS ONE. Congratulations! Your manuscript is now with our production department. 

With kind regards,

on behalf of

Dr. Mukesh Dhamala 

Academic Editor

PLOS ONE